# Comparing the Safety and Efficacy of Microwave Ablation Using Thermosphere^TM^ Technology versus Radiofrequency Ablation for Hepatocellular Carcinoma: A Propensity Score-Matched Analysis

**DOI:** 10.3390/cancers13061295

**Published:** 2021-03-15

**Authors:** Hidekatsu Kuroda, Tomoaki Nagasawa, Yudai Fujiwara, Hiroki Sato, Tamami Abe, Yohei Kooka, Kei Endo, Takayoshi Oikawa, Kei Sawara, Yasuhiro Takikawa

**Affiliations:** Division of Hepatology, Department of Internal Medicine, Iwate Medical University, Nishitokuta 2-1-1, Yahaba-cho, Shiwa-gun, Iwate 028-3694, Japan; tomoakin@iwate-med.ac.jp (T.N.); yudfuji@iwate-med.ac.jp (Y.F.); hsato@iwate-med.ac.jp (H.S.); tabe@iwate-med.ac.jp (T.A.); ykooka@iwate-med.ac.jp (Y.K.); keiendo@iwate-med.ac.jp (K.E.); tyoikawa@iwate-med.ac.jp (T.O.); ksawara@iwate-med.ac.jp (K.S.); ytakikaw@iwate-med.ac.jp (Y.T.)

**Keywords:** hepatocellular carcinoma, microwave ablation, radiofrequency ablation

## Abstract

**Simple Summary:**

Microwave ablation using Thermosphere^TM^ technology is a novel locoregional treatment for hepatocellular carcinoma. This study compared the safety and efficacy outcomes of this microwave ablation strategy versus radiofrequency ablation using propensity score-matched analysis. Microwave ablation led to a high rate of curative ablation (94.7%) and a low rate of local recurrence (3.3%), with an overall survival rate of 99.3% at 1 year (recurrence-free survival: 81.1%) and 88.4% at 2 years (recurrence-free survival: 60.5%). There were no significant differences in survival outcomes after microwave and radiofrequency ablation. However, microwave ablation required significantly fewer insertions (1.22 ± 0.49 vs. 1.59 ± 0.94; *p* < 0.0001). Based on the similar survival outcomes, we recommend microwave ablation using Thermosphere^TM^ technology for hepatocellular carcinoma with a diameter of >2 cm because of the lower number of insertions.

**Abstract:**

There is limited information regarding the oncological benefits of microwave ablation using Thermosphere^TM^ technology for hepatocellular carcinoma. This study compared the overall survival and recurrence-free survival outcomes among patients with hepatocellular carcinoma after microwave ablation using Thermosphere^TM^ technology and after radiofrequency ablation. Between December 2017 and August 2020, 410 patients with hepatocellular carcinoma (a single lesion that was ≤5 cm or ≤3 lesions that were ≤3 cm) underwent ablation at our institution. Propensity score matching identified 150 matched pairs of patients with well-balanced characteristics. The microwave ablation and radiofrequency ablation groups had similar overall survival rates at 1 year (99.3% vs. 98.2%) and at 2 years (88.4% vs. 87.5%) (*p* = 0.728), as well as similar recurrence-free survival rates at 1 year (81.1% vs. 76.2%) and at 2 years (60.5% vs. 62.1%) (*p* = 0.492). However, the microwave ablation group had a significantly lower mean number of total insertions (1.22 ± 0.49 vs. 1.59 ± 0.94; *p* < 0.0001). This retrospective study revealed no significant differences in the overall survival and recurrence-free survival outcomes after microwave ablation or radiofrequency ablation. However, we recommend microwave ablation for hepatocellular carcinoma tumors with a diameter of >2 cm based on the lower number of insertions.

## 1. Introduction

Hepatocellular carcinoma (HCC) is the most common primary malignancy of the liver, the fifth most common cancer type among men and the seventh most common cancer type among women [1]. The worldwide incidence of HCC is 10.1 cases per 100,000 person-years [2]. The most commonly used staging system is the modified Barcelona Clinic Liver Cancer (BCLC) staging system [3] and surgical resection or ablation is recommended for very early and early HCC (BCLC stage 0 and stage A). Surgical treatment can be curative at these stages, though most HCC patients are not eligible for surgical resection because they typically present with advanced disease and underlying liver dysfunction [4,5]. Thus, locoregional therapies such as radiofrequency ablation (RFA) are recommended as first-line treatment for small and single tumors in the guidelines from the American Association for the Study of Liver Diseases, the European Association for the Study of the Liver and the Asian Pacific Association for the Study of the Liver [6,7,8].

There are several types of ablation therapy, including RFA, which is a minimally invasive locoregional treatment. Studies have indicated that RFA provides 5-year survival rates of 39.9–68.5% and local tumor progression rates of 2.4–27.0% [9,10,11,12,13,14]. However, HCC nodules are often incompletely ablated because of the heat sink effect created by large peritumoral blood vessels. Furthermore, 10–25% of HCC patients may not be eligible for RFA [15,16]. Microwave ablation (MWA) was developed several years after RFA had become established as the nonsurgical standard of care for early HCC [17]. Conventional MWA provides theoretical advantages over RFA, including higher ablative temperatures and a lesser heat sink effect, although the largest phase II randomized controlled trial comparing MWA and RFA (152 patients) did not identify significant differences in the 2-year rate of local tumor progression [18]. Moreover, Chong et al. reported that MWA was no different from RFA with respect to ablation completeness and survival rates in a randomized controlled study [19]. These findings may be related to first-generation MWA strategies having ablation zones with unpredictable sizes and shapes.

Recent improvements in MWA technology have been able to provide large and spherical treatment areas. For example, a high-powered (2.4 GHz) single-antenna MWA system has been developed using Thermosphere^TM^ technology (Emprint^TM^ system; Covidien, Boulder, CO, USA) and has been used clinically [20,21,22]. Thermosphere^TM^ technology provides thermal, field and wavelength control, which allows the system to produce reliable and large spherical ablation zones [20]. Furthermore, this system provides theoretical benefits by overcoming issues with tissue charring and the heat sink effect [21,22]. However, we are not aware of any definitive evidence that supports the superiority of MWA using Thermosphere^TM^ technology over RFA. Therefore, this study evaluated safety, overall survival (OS) and recurrence-free survival (RFS) outcomes after HCC treatment with MWA using Thermosphere^TM^ technology or RFA. Our hypothesis was that there would be no significant differences in the survival outcomes of the two groups and we performed propensity score matching (PSM) to minimize potential confounding effects caused by differences in the patients’ characteristics.

## 2. Materials and Methods

### 2.1. Patients

This retrospective study evaluated consecutive patients with complete pre-ablation and post-ablation information in our institutional database. Between December 2017 and August 2020, we identified 410 patients who underwent MWA using Thermosphere^TM^ technology or RFA for HCC at Iwate Medical University Hospital. The diagnosis of HCC was based on findings from tumor-targeted biopsy, ultrasonography, computed tomography (CT) and magnetic resonance imaging. Patients had been considered eligible for MWA or RFA based on the following: (1) HCC nodules that were unresectable or the patient had refused surgery, (2) a single nodule with a diameter of ≤5 cm or ≤3 nodules with diameters of ≤3 cm, (3) Child–Pugh grade A–B disease, (4) platelet count of ≥ 5.0 × 10^5^/mm^3^, (5) total bilirubin concentration of <3.0 mg/dL and (6) prothrombin activity of ≥50%. If present, ascites was controlled before the intervention using diuretics. The exclusion criteria were: (1) patients with previous or simultaneous malignancies, (2) portal vein tumor-related thrombosis and (3) extrahepatic metastasis. The decision to perform MWA or RFA was based on the consensus of at least two hepatologists (HK and TO) who each had >15 years of experience. In most cases, there was no disagreement between them; in a few cases involving differing opinions, a third hepatologist (KE) made the final decision. In the early stages of this study, we had more experience with RFA than with MWA. Therefore, MWA for HCC nodules at high-risk locations was initially chosen more carefully.

We defined high-risk HCC nodule locations as being adjacent to large vessels or extrahepatic organs, based on a previous report by Teratani et al. [23]. These locations were considered to be located <5 mm from a first or second branch of the portal vein, the base of the hepatic veins, the inferior vena cava, the heart, the lungs, the gallbladder, the right kidney, or the intestinal tract. In total, 405 patients were considered eligible, including 150 patients who underwent MWA and 255 patients who underwent RFA. We used PSM (1:1 ratio) to create two groups of matched patients (*n* = 150 each) who underwent MWA or RFA.

The retrospective study protocol was approved by the ethics committee of Iwate Medical University on 7 November 2018 (MH2018-557). Patients had provided their written informed consent for the original procedures in accordance with the principles of the Declaration of Helsinki (as revised in Fortaleza, 2013).

### 2.2. MWA Procedure

The MWA procedure using Thermosphere^TM^ technology was performed under real-time ultrasound guidance using the 2.4-GHz MWA system generator (Emprint^TM^ system; Covidien, Boulder, CO, USA) and a 15–20 cm 13-gauge saline-cooled coaxial antenna. When a nodule was judged to be in a high-risk location (defined in Section 2.1), the route of electrode insertion was carefully selected based on ultrasonography findings to avoid injury to the lungs, intestinal tract, gall bladder, portal vein, inferior vena cava and bile duct. We also used an artificial ascites or pleural effusion technique to visualize the entire nodule when it was located adjacent to the intestinal tract or diaphragm. Fusion imaging or contrast-enhanced ultrasonography was used if required during the MWA procedure to improve nodule detection and localization. In cases with hypervascularity and tumor size of ≥3 cm, MWA was combined with transcatheter arterial chemoembolization (TACE).

After induction of local anesthesia, the MWA antenna was inserted under ultrasound guidance and introduction into the tumor was confirmed. Intravenous midazolam administration (0.06 mg/kg) was used to achieve sedation during the ablation. A tip temperature of <20 °C was maintained using a peristaltic pump and chilled saline solution. According to the manufacturer’s specifications, separate MWA sessions were performed at 45 W (1 min), 60 W (1 min), 75 W (1 min) and 100 W (3.5–8.5 min) to achieve an optimal necrosis volume, which was based on the nodule diameters that had been calculated using the preprocedural CT findings. During the ablation, a thermocouple embedded in the electrode tip continuously monitored the local temperature and tissue impedance was also continuously monitored using circuitry that was incorporated in the generator. After completion of the MWA procedure, intravenous flumazenil administration (0.5 mg) was used to induce recovery from sedation. Antibiotic treatment was administered on the day of the procedure and the next day, with continued treatment for patients who had a fever.

### 2.3. RFA Procedure

The RFA procedure was performed using a 17-gauge internally cooled length-adjustable electrode (Proteus^®^ RF Electrode; STARmed, Gyeonggi-do, Republic of Korea) with a 200 W RF generator (VIVA RF System; STARmed, Gyeonggi-do, Republic of Korea). The length of the active tip was selected based on the tumor’s size. After insertion of the electrode into the lesion, we started ablation at 60 W for 3 cm exposed tips or 40 W for 2 cm exposed tips and the power was increased to 120 W at a rate of 20 W/min. When a rapid increase in impedance was observed during thermal ablation, we minimized the output for 15 s and restarted the emission at a lower output. The procedures for artificial ascites or pleural effusion techniques, fusion imaging, contrast-enhanced ultrasonography, combination with TACE, sedatives and antibiotic treatment were the same as the procedures described for MWA.

### 2.4. Assessing Treatment Efficacy and Follow-Up

Dynamic CT (section thickness: 5 mm) was performed 1–3 days after the MWA or RFA session to evaluate the treatment’s efficacy. The radicality of MWA or RFA treatment was classified into four grades (R grades: A, B, C and D) as previously reported by Nishikawa et al. based on the extent of the resected tumor margin [24]. Grade A (absolutely curative) was defined as an ablative margin of ≥5 mm around the entire tumor. Grade B (relatively curative) was defined as an ablative margin that extended around the entire tumor with a margin of <5 mm in some places. Grade C (relatively non-curative) was defined as an incomplete ablative margin despite no apparent residual tumor. Grade D (absolutely non-curative) was defined as apparent incomplete tumor ablation. Patients received additional ablation sessions as much as possible to achieve an R grade A–B response. Follow-up consisted of monthly blood tests and tumor marker monitoring at the outpatient clinic; ultrasonography and dynamic CT were performed every 3 months. Intrahepatic HCC recurrence was classified as either tumor recurrence at a site distant from the primary tumor or adjacent to the treated site (local tumor progression). If the patient fulfilled the original eligibility criteria, MWA or RFA was performed for recurrent HCC tumors.

### 2.5. PSM Analysis

We performed PSM to decrease the effects of selection bias on the survival analyses by creating matched groups of patients who had undergone MWA or RFA. The propensity score model included age, sex, performance status (PS) score, etiology, naive or non-naive status, Child-Pugh grade, serum α-fetoprotein (AFP) concentration, des-γ-carboxy prothrombin (DCP) concentration, tumor size, tumor number, TACE before ablation and high-risk locations. The propensity scores were calculated by applying these variables to a logistic regression model and C-statistics were calculated to evaluate the goodness of fit. One-to-one PSM was performed using a caliper width of <0.2 of the pooled standard deviation of the estimated propensity scores. PSM was performed using SPSS software (version 23; IBM Corp., Armonk, NY, USA).

### 2.6. Statistical Analysis

All statistical analyses were performed using SPSS software (version 23.0) or XLSTAT 2020 software (Microsoft Corp., Redmond, WA, USA). Continuous variables were presented as the mean ± standard deviation or median (interquartile range) and were analyzed using the Student’s t test or Mann–Whitney U test. Categorical variables were presented as the number (percentage) and were analyzed using the Pearson’s χ^2^ test or Fisher’s exact test. The OS and RFS curves were created using the Kaplan–Meier method and compared using the log-rank test. Independent predictors of OS and RFS were evaluated using multivariable Cox proportional hazard regression analyses, which were adjusted for factors with a *p*-value of ≤0.05 in the univariate analyses. Differences were considered statistically significant at *p*-values of <0.05.

## 3. Results

### 3.1. Baseline Patient Characteristics

During the study period, 410 patients underwent MWA or RFA, although 6 patients were excluded because of loss to follow-up (*n* = 4) or cancer in other organs (*n* = 2). Thus, 404 patients were divided into the MWA group (*n* = 150, 37.1%) and the RFA group (*n* = 254, 62.9%; Figure 1). The characteristics of the unmatched groups are shown in Table 1 and we observed substantial differences in some of the groups’ characteristics. Relative to the RFA group, the MWA group had a significantly larger tumor size (*p* < 0.001) and a higher proportion of tumors in high-risk locations (*p* < 0.039). The median follow-up time for all patients was 405.5 days (MWA group: 415.7 days, RFA group: 399.5 days). No significant differences were observed in OS rates between the MWA and RFA groups at 1 year (99.3% vs. 99.2%) or at 2 years (88.2% vs. 81.6%, *p* = 0.169; Figure 2A). Furthermore, no significant differences were observed in RFS rates between the MWA and RFA groups at 1 year (81.1% vs. 73.8%) or at 2 years (60.5% vs. 54.6%, *p* = 0.151; Figure 2B).

### 3.2. Patient Characteristics in the PSM Cohort

The PSM analysis identified 150 matched pairs of patients from each group. The matched groups of patients had similar baseline characteristics (Table 1), including age, cause of underlying liver disease, background liver function, tumor size and proportion of tumors in high-risk locations (all *p* > 0.05). The median follow-up time for the PSM cohort was 406.2 days (MWA group: 415.7 days, RFA group: 391.5 days). The matched MWA and RFA groups had similar OS rates at 1 year (99.3% vs. 98.2%) and at 2 years (88.4% vs. 87.5%, *p* = 0.728; Figure 3B). Furthermore, the matched MWA and RFA groups had similar RFS rates at 1 year (81.1% vs. 76.2%) and at 2 years (60.5% vs. 62.1%, *p* = 0.492; Figure 3B).

### 3.3. Treatment Efficacy in the PSM Cohort

In the MWA group, the treatment outcomes after the first session were R grade A (61 cases), grade B (67 cases), grade C (20 cases) and grade D (two cases). Similarly, the treatment outcomes after the first session in the RFA group were R grade A (53 cases), grade B (76 cases), grade C (18 cases) and grade D (3 cases). The differences between the two groups were not statistically significant (*p* = 0.611; Table 2). The final treatment outcomes in the MWA group were R grade A (82 cases), grade B (60 cases), grade C (six cases) and grade D (two cases). The final treatment outcomes in the RFA group were grade A (80 cases), grade B (57 cases), grade C (10 cases) and grade D (three cases). The differences between the two groups were also not statistically significant (*p* = 0.729).

The total numbers of sessions were 1.14 ± 0.34 for the MWA group and 1.21 ± 0.42 for the RFA group (*p* = 0.477). However, the MWA group required significantly fewer total insertions (1.22 ± 0.49 vs. 1.59 ± 0.94; *p* < 0.0001). Figure 4 shows the total number of insertions required until the end of treatment according to tumor size. In the MWA group, the total numbers of insertions were 1 for <10-mm tumors, 1.05 ± 0.22 for 10–19 mm tumors, 1.13 ± 0.34 for 20–29 mm tumors, 1.22 ± 0.42 for 30–39 mm tumors and 2.00 ± 0.77 for 40–50 mm tumors (*p* < 0.0001). Similarly, in the RFA group, the total numbers of insertions were 1 for <10 mm tumors, 1.14 ± 0.39 for 10–19 mm tumors, 1.48 ± 0.57 for 20–29 mm tumors, 2.94 ± 1.03 for 30–39 mm tumors and 3.83 ± 1.47 times for 40–50 mm tumors (*p* < 0.0001). When we compared the MWA and RFA groups, there was no significant difference in the total number of insertions for tumors that were <19 mm, although the MWA group required significantly fewer insertions for tumors that were >20 mm (*p* < 0.001).

### 3.4. Complications and Recurrences in the PSM Cohort

During the follow-up period, 19 complications were observed (6.3% of patients, 5.4% of sessions; Table 2), although there were no complication-related deaths. Complications were experienced by 8 patients (5.3%) in the MWA group, which included hepatic infarction (3 cases), intraperitoneal hemorrhage (2 cases), pleural effusion (1 case), hepatic abscess (1 case) and portal vein thrombosis (1 case). Complications were also experienced by 11 patients (7.3%) in the RFA group, which included hepatic infarction (2 cases), intraperitoneal hemorrhage (1 case), pleural effusion (1 case), bile peritonitis (1 case), duodenal perforation (1 case), hepatic abscess (1 case), neoplastic seeding (1 case), skin burn (1 case), biloma (1 case) and portal vein thrombosis (1 case). There was no significant inter-group difference in terms of complications (*p* = 0.477).

Tumor recurrence was identified in 67 of 300 patients during the follow-up period. The median time to tumor recurrence was 294 days (range: 30–821 days). The MWA group included 29 patients with intrahepatic distant recurrence and 5 patients with local tumor progression. The RFA group included 26 patients with intrahepatic distant recurrence and 7 patients with local tumor progression. There was no significant inter-group difference in terms of recurrence (*p* = 0.814; Table 2).

### 3.5. Univariate and Multivariable Analyses of OS and RFS in the PSM Cohort

Univariate analyses revealed that OS was significantly associated with a final outcome of R grade C–D (*p* = 0.001), Child–Pugh grade B disease (*p* = 0.002) and non-naive status (*p* = 0.014). The multivariable analyses revealed that OS was independently associated with a final outcome of R grade C–D (hazard ratio [HR]: 5.143, 95% confidence interval [CI]: 2.087–9.652, *p* = 0.001) and Child–Pugh grade B disease (HR: 3.734, 95% CI: 1.190–8.079, *p* = 0.003; Table 3).

The univariate analyses revealed that RFS was significantly associated with a final outcome of R grade C–D (*p* < 0.001), Child–Pugh grade B disease (*p* = 0.009), an AFP concentration of >100 ng/mL (*p* = 0.034), tumor size (*p* = 0.025) and the number of tumors (*p* = 0.032). The multivariable analyses revealed that RFS was independently predicted by a final outcome of R grade C–D (HR: 8.837, 95% CI: 4.563–17.115, *p* < 0.001), Child–Pugh grade B disease (HR: 2.459, 95% CI: 1.307–4.628, *p* = 0.005), an AFP concentration of >100 ng/mL (HR: 2.005, 95% CI: 1.065–3.775, *p* = 0.031) and the number of tumors (HR: 1.406, 95% CI: 1.108–1.865, *p* = 0.039; Table 4). The OS and RFS outcomes were not independently associated with MWA or RFA as the ablation modality.

## 4. Discussion

This retrospective PSM-based study revealed no significant differences in OS and RFS outcomes between groups of patients who underwent MWA using Thermosphere^TM^ technology or RFA. Moreover, there were no statistically significant differences in terms of the R grade treatment outcome, total number of sessions, total number of complications, or local tumor progression. These results confirm that MWA is a safe and effective strategy for locoregional treatment of HCC, similar to RFA.

The MWA strategy is based on dielectric heating that occurs when an imperfect dielectric material is exposed to an alternating electromagnetic field [25]. A microwave field oscillates rapidly and rotates polar molecules (primarily water) so that they oscillate out of phase, which causes some electromagnetic energy to be converted into heat. Several studies have compared the efficacy of first-generation MWA to that of RFA, which revealed that MWA had comparable safety and long-term efficacy for liver tumors with a diameter of <4 cm [18,19,26]. New MWA technologies have also been developed and attracted significant interest regarding their safety and efficacy, although there are few reports regarding new MWA technologies as appropriate treatment strategies [20,21,22,27]. To the best of our knowledge, this is the first study to evaluate the short-term and medium-term survival, efficacy and safety of MWA.

The reported advantages of MWA include higher intratumoral temperatures, larger and more predictable ablation volumes, faster ablation, less procedural pain and the ability to use multiple applicators [25,28,29,30]. To provide these advantages, Thermosphere^TM^ technology relies on thermal, field and wavelength control [20]. Thermal control is achieved via internal cooling of the probe and cables using a sterile saline solution, which is circulated down the shaft to the distal probe tip and this process ensures a reliable ablation zone that remains unaffected by tissue desiccation near the shaft. Field control is based on the ability to confirm electron movements in the probe, which produces the desired field shape and ensures that it remains constant despite ablation-related changes in the tissue environment. Heating of the surrounding tissue also influences the tissue properties and alters the dielectric constant, while wavelength control minimizes changes in the dielectric constant immediately around the probe. Therefore, these three levels of control allow MWA to produce reliable and large spherical ablation zones.

This retrospective study revealed no significant differences in terms of OS or RFS when MWA or RFA was used to treat patients with HCC. In addition, the OS and RFS outcomes were independently predicted by the final R grade of treatment response and the Child-Pugh disease classification, whereas the outcomes were not predicted by the ablation modality (MWA or RFA). The curative ablations rates were high in both the MWA group (142/150, 94.7%) and in the RFA group (137/150, 91.3%), whereas the rates of local recurrence were low in the MWA group (5/150, 3.3%) and in the RFA group (7/150, 4.7%). Therefore, we speculate that the ablation modality would not significantly influence OS and RFS outcomes as long as the margins are secure. However, curative ablation using MWA required fewer insertions (vs. RFA) and this difference was significant for tumors with a maximum diameter of >2 cm. Thus, the smaller number of antenna insertions makes MWA an attractive locoregional treatment for HCC. Moreover, the ablation area of MWA can be precisely controlled based on time and output power [21,22], which permits a single MWA antenna to treat small nodules (<1 cm) and larger nodules up to a size of 4 cm.

There are concerns regarding bile duct injury related to the higher temperature during MWA and also concerns regarding bleeding that is related to the thicker antennas. In the present study, 8 patients in the MWA group (5.3%) experienced complications. However, there were no complication-related deaths and all patients recovered after conservative treatment. Furthermore, there were no significant differences in the complications between the MWA and RFA groups.

This study has several limitations. First, this is a retrospective cohort study and not a randomized controlled trial. We used PSM and multiple Cox regression analyses to minimize the bias effects of group differences. Second, the follow-up time was short and prolonged follow-up will be needed to collect additional data regarding OS and RFS outcomes. Third, this study was performed at a single center and larger prospective multicenter studies are needed to validate our findings.

## 5. Conclusions

This study revealed no significant differences in terms of OS and RFS outcomes among patients who underwent MWA using Thermosphere^TM^ or RFA for HCC. However, based on the smaller number of insertions, we recommend MWA for HCC tumors with a diameter of >2 cm.

## Figures and Tables

**Figure 1 cancers-13-01295-f001:**
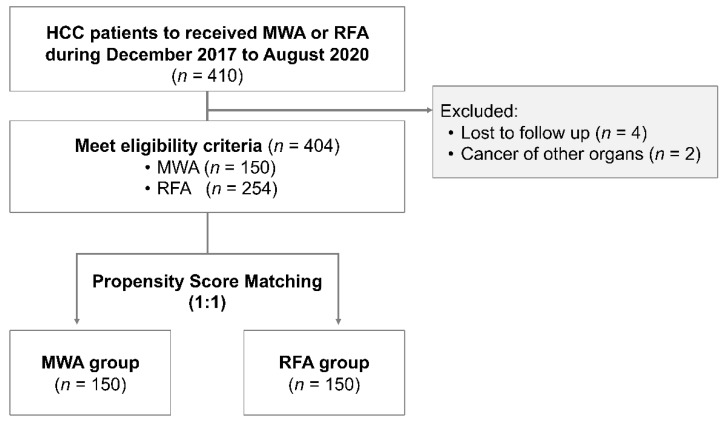
Study flowchart. MWA, microwave ablation; RFA, radiofrequency ablation.

**Figure 2 cancers-13-01295-f002:**
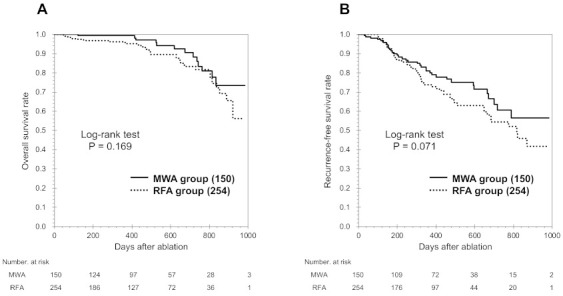
Kaplan-Meier curves for overall survival (OS, **A**) and recurrence-free survival (RFS, **B**) among all patients. No significant differences were observed between the groups that underwent microwave ablation (MWA) and radiofrequency ablation (RFA).

**Figure 3 cancers-13-01295-f003:**
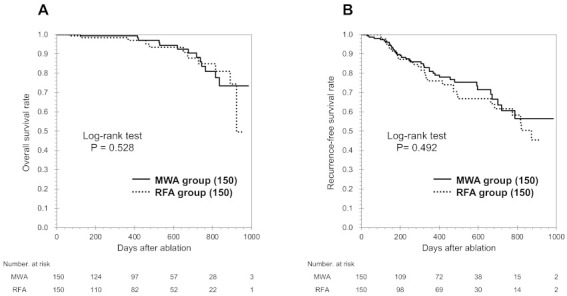
Kaplan-Meier survival curves for overall survival (OS, **A**) and recurrence-free survival (RFS, **B**) in the PSM cohort. No significant differences were observed between the groups that underwent microwave ablation (MWA) and radiofrequency ablation (RFA).

**Figure 4 cancers-13-01295-f004:**
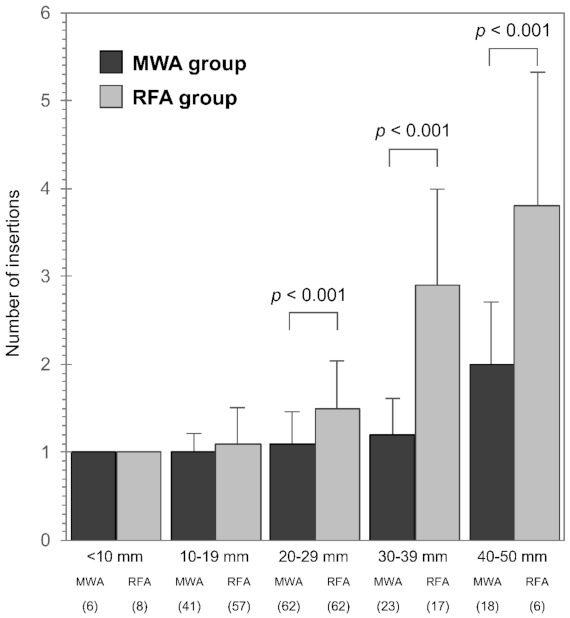
Total number of insertions according to tumor size. There was no significant difference in the total number of insertions for tumors that were sized <19 mm, although significantly fewer insertions were required for tumor sizes of >20 mm in the MWA group (*p* < 0.001). MWA, microwave ablation; RFA, radiofrequency ablation.

**Table 1 cancers-13-01295-t001:** Comparison of the MWA and RFA groups in the unmatched and propensity score-matched cohorts.

Characteristic	Unmatched Cohort			Propensity Score-Matched Cohort		
	MWA (*n* = 150)	RFA (*n* = 254)	*p*-Value	MWA (*n* = 150)	RFA (*n* = 150)	*p*-Value
Sex (male/female)	109/41	187/67	0.837	109/41	112/38	0.694
Mean age (years)	71.6 ± 9.10	71.8 ± 10.3	0.605	71.6 ± 9.10	72.3 ± 9.60	0.369
PS score (0/1)	140/10	236/18	0.901	140/10	138/12	0.658
Etiology (HBV/HCV/Alcohol/Others)	18/62/34/36	37/132/55/30	0.195	18/62/34/36	24/64/33/29	0.689
Naive/non-naive	73/77	143/111	0.137	73/77	75/75	0.817
Child-Pugh grade (A/B)	132/18	208/46	0.104	132/18	131/19	0.861
T.Bil (mg/dL)	0.6 [0.5–0.9]	0.6 [0.5–0.9]	0.888	0.6 [0.5–0.9]	0.6 [0.5–0.8]	0.798
Alb (g/L)	3.8 [3.5–4.1]	3.8 [3.4–4.2]	0.942	3.8 [3.5–4.1]	3.8 [3.5–4.1]	0.684
AST (U/L)	56.8 [31.5–64.1]	54.3 [30.5–67.1]	0.596	56.8 [31.5–64.1]	55.1 [30.9–68.3]	0.605
PT (%)	86.0 [74.0–95.0]	84.5 [72.7–95.6]	0.814	86.0 [74.0–95.0]	88.5 [75.0–97.0]	0.752
Plt (×10^4^/mm^3^)	11.7 [7.5–14.1]	11.8 [7.0–15.9]	0.851	11.7 [7.5–14.1]	11.9 [7.1–14.9]	0.765
AFP > 100 ng/mL (%)	16.0% (24/150)	14.2% (36/254)	0.618	16.0% (24/150)	14.0% (21/150)	0.698
DCP > 40 mAU/mL (%)	35.3% (53/150)	38.5% (98/254)	0.238	35.3% (53/150)	36.7% (55/150)	0.759
Tumor size (mm)	26.8 ± 11.3 *	20.3 ± 8.60	< 0.001	26.8 ± 11.3	24.6 ± 10.1	0.174
Number of tumors	1.32 ± 0.54	1.48 ± 0.85	0.082	1.32 ± 0.54	1.39 ± 0.76	0.926
High-risk locations met (%)	26.0% (39/150) **	35.4% (90/254)	0.039	26.0% (39/150)	22.7% (34/150)	0.275
TACE before ablation (%)	37.3% (56/150)	35.8% (91/254)	0.761	37.3% (56/150)	32.7% (49/150)	0.432

The values represent the mean ± standard deviation, the median [25th–75th percentile], or the number of patients. * *p* < 0.01 (compared to RFA), ** *p* < 0.05 (compared to RFA). Abbreviations: MWA, microwave ablation; RFA, radiofrequency ablation; PS, performance status; HBV, hepatitis B virus; HCV, hepatitis C virus; NAFLD, non-alcoholic fatty liver disease; T.Bil, total bilirubin; Alb, albumin; AST, aspartate aminotransferase; PT, prothrombin time; Plt, platelet; AFP, α-fetoprotein; DCP, des-γ-carboxy prothrombin; TACE, transarterial chemoembolization.

**Table 2 cancers-13-01295-t002:** Treatment parameters and outcomes in the propensity score-matched cohort.

Variables	MWA (*n* = 150)	RFA (*n* = 150)	*p*-Value
Treatment outcome after 1 session (R grade A/B/C/D)	61/67/20/2	53/76/18/3	0.611
Final treatment outcome (R grade A/B/C/D)	82/60/6/2	80/57/10/3	0.729
Total number of ablation sessions	1.14 ± 0.34	1.21 ± 0.42	0.123
Total number of insertions	1.22 ± 0.49	1.59 ± 0.94	<0.0001
Total number of complications	5.3% (8/150)	7.3% (11/150)	0.477
Hepatic infarction	3	2	
Intraperitoneal hemorrhage requiring blood transfusion	2	1	
Pleural effusion requiring drainage	1	1	
Bile peritonitis	0	1	
Duodenal perforation	0	1	
Hepatic abscess requiring drainage	1	1	
Neoplastic seeding	0	1	
Skin burn	0	1	
Biloma	0	1	
Portal vein thrombosis	1	1	
Recurrences	22.7% (34/150)	22.0% (33/150)	0.814
Intrahepatic distant recurrence	29	26	
Local tumor progression	5	7	

Data are presented as mean ± standard deviation or percentage (fraction). Abbreviations: MWA, microwave ablation; RFA, radiofrequency ablation; R grade, radicality grade.

**Table 3 cancers-13-01295-t003:** Univariate and multivariable analyses of overall survival in the propensity score-matched cohort.

Parameter	Univariate Analysis			Multivariate Analysis		
	HR (95% CI)		*p*-Value	HR (95% CI)		*p*-Value
Sex (male)	0.979	(0.937–1.023)	0.349			
Age (>70)	0.885	(0.366–2.137)	0.785			
PS score (1)	1.132	(0.601–2.121)	0.376			
Etiology (HBV + HCV)	0.602	(0.269–1.347)	0.217			
Non-naive	2.152	(1.236–6.583)	**0.014**	1.322	(0.978–5.902)	0.057
Child-Pugh (grade B)	2.855	(1.262–6.261)	**0.002**	3.734	(1.190–8.079)	**0.003**
T.Bil (>1.0 mg/dL)	1.925	(0.808–4.584)	0.139			
Alb (>3.5 g/L)	0.880	(0.373–2.080)	0.771			
AST (>50 U/L)	0.723	(0.457–2.258)	0.852			
PT (>80 %)	0.995	(0.970–1.021)	0.714			
Plt (<10 ×10^4^/mm^3^)	1.443	(0.966–2.157)	0.098			
AFP (>100 ng/mL)	1.682	(0.563–4.003)	0.348			
DCP (>40 mAU/mL)	1.136	(0.466–2.768)	0.802			
Tumor size (>25 mm)	1.027	(0.993–1.061)	0.116			
Number of tumors (multiple)	1.102	(0.338–1.459)	0.343			
High-risk tumor location (yes)	0.301	(0.071–1.284)	0.105			
TACE before ablation (yes)	1.230	(0.537–2.715)	0.625			
Ablation modality (MWA)	0.867	(0.386–1.944)	0.729			
Final R grade (C–D)	4.803	(1.896–9.622)	**0.001**	5.143	(2.087–9.652)	**0.001**

Significant differences are indicated using bold *p*-values. Abbreviations: HR, hazard ratio; 95% CI, 95% confidence interval; PS, performance status; HBV, hepatitis B virus; HCV, hepatitis C virus; T.Bil, total bilirubin; Alb, albumin; AST, aspartate aminotransferase; PT, prothrombin time; Plt, platelet count; AFP, α-fetoprotein; DCP, des-γ-carboxy prothrombin; TACE, transarterial chemoembolization; MWA; microwave ablation; R grade, radicality grade.

**Table 4 cancers-13-01295-t004:** Univariate and multivariate analysis for recurrence-free survival in the propensity score-matched cohort.

Parameter	Univariate Analysis			Multivariate Analysis		
	HR (95% CI)		*p*-Value	HR (95% CI)		*p*-Value
Sex (male)	0.985	(0.960–1.010)	0.235			
Age (>70)	1.410	(0.795–2.501)	0.239			
PS score (1)	1.231	(0.701–2.365)	0.321			
Etiology (HBV+HCV)	1.298	(0.777–2.169)	0.319			
Non-naive	1.171	(0.722–1.901)	0.522			
Child-Pugh (grade B)	2.243	(1.201–4.189)	**0.009**	2.459	(1.307–4.628)	**0.005**
T.Bil (>1.0 mg/dL)	1.977	(0.835–3.442)	0.216			
Alb (>3.5 g/L)	0.537	(0.254–1.587)	0.314			
AST (>50 U/L)	1.052	(0.784–1.412)	0.734			
PT (>80 %)	1.004	(0.989–1.019)	0.631			
Plt (<10 ×10^4^/mm^3^)	0.991	(0.973–1.009)	0.332			
AFP (>100 ng/mL)	1.967	(1.052–3.672)	**0.034**	2.005	(1.065–3.775)	**0.031**
DCP (>40 mAU/mL)	1.237	(0.995–1.813)	0.067			
Tumor size (>25 mm)	1.215	(1.032–2.953)	**0.025**	1.015	(0.992–3.022)	0.275
Number of tumors (multiple)	1.182	(1.012–2.672)	**0.032**	1.406	(1.108–1.865)	**0.039**
High-risk tumor location (yes)	0.533	(0.279–1.017)	0.056			
TACE before ablation (yes)	1.473	(0.911–2.382)	0.114			
Ablation modality (MWA)	0.846	(0.526–1.361)	0.491			
Final R grade (C–D)	9.463	(5.329–16.780)	**<0.001**	8.837	(4.563–17.115)	**<0.001**

Significant differences are indicated using bold *p*-values. Abbreviations: HR, hazard ratio; 95% CI, 95% confidence interval; PS, performance status; HBV, hepatitis B virus; HCV, hepatitis C virus; T.Bil, total bilirubin; Alb, albumin; AST, aspartate aminotransferase; PT, prothrombin time; Plt, platelet count; AFP, α-fetoprotein; DCP, des-γ-carboxy prothrombin; TACE, transarterial chemoembolization; MWA; microwave ablation; R grade, radicality grade.

## Data Availability

The data presented in this study are available in the manuscript and its associated files.

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
