# Peer review of "Comparing the Safety and Efficacy of Microwave Ablation Using ThermosphereTM Technology versus Radiofrequency Ablation for Hepatocellular Carcinoma: A Propensity Score-Matched Analysis"

_cancers, 2021, doi:10.3390/cancers13061295_

Round 1

Reviewer 1 Report

Remarks to the Authors:

This retrospective study is a comparison of the outcomes of MWA treatment utilising Thermosphere technology and the outcomes of RFA treatment in 410 HCC patients. The study showed that there were no significant differences in overall survival or recurrence-free survival between patients treated with MWA or RFA. However, the MWA group showed a significantly lower number of insertions for tumours larger than 2 cm. To avoid selection bias, a propensity score-matched analysis was used to find 150 matched pairs of patients.

Overall the work is carefully done and the conclusions look sound. The Study shows a comprehensive overview about the used technique and the outcome of the treatment. However, the brand name Thermosphere is mentioned often in this text, so as not to create a false impression in the reader's mind, it might be preferable to reduce the mention of the brand name.

I have only few minor comments.

Minor comments:

Page 2 - row 45:

Which classification system is meant by Stage 0 and Stage A?

Page 5 - Table1:

In Table 1, errors have occurred in the total numbers, e.g. for sex, PS score or Child-Pugh grade. The number after the "/" always has one zero extra. Please also check whether the error has moved into the P value calculation.

Page 6 - figures (count also for figure 4 on page 8):

It is praiseworthy that real data are displayed under the figures, however, for the comprehensibility it would be good if there were a note for it in the figures caption.

Page 9 - table 3 and 4:

The table description states that significant P values are printed in bold, but there are no P values in bold in the table.

Page 9 - row 274

Naive instead of naïve

Author Response

RESPONSES TO REVIEWER 1

We thank the reviewer for the thoughtful and constructive review of our manuscript. We have addressed all of the reviewer's concerns, and our detailed responses to the reviewer’s comments are as follows:

Major points

1) The brand name Thermosphere is mentioned often in this text, so as not to create a false impression in the reader's mind, it might be preferable to reduce the mention of the brand name.

Response:

We thank the reviewer for the suggestion regarding this point. In accordance with the reviewer's valuable advice, we have deleted some instances of the brand name to avoid repetition (Materials and Methods: Page 2, line 90. Page 3, line 107. Discussion: Page 11 line 314, 326, 339, 351, and 352. Conclusions: Page 12 line 370).

Minor points

1) Page 2 - row 45: Which classification system is meant by Stage 0 and Stage A?

Response:

We used the Barcelona Clinic Liver Cancer (BCLC) staging system. We agree that the classification system should be specified. Accordingly, we have added this information to the text (Introduction: Page 2, line 47).

2) Page 5 - Table1: In Table 1, errors have occurred in the total numbers, e.g. for sex, PS score or Child-Pugh grade. The number after the "/" always has one zero extra. Please also check whether the error has moved into the P value calculation.

Response:

We thank the reviewer for this comment. We have checked the statistical data and revised Table 1 (Page 5-6) accordingly.

3) Page 6 - figures (count also for figure 4 on page 8): It is praiseworthy that real data are displayed under the figures, however, for the comprehensibility it would be good if there were a note for it in the figures caption.

Response:

We appreciate the reviewer’s comment. Based on this suggestion, we have revised Figures 2 and 3 (Page 6-7).

4) Page 9 - table 3 and 4: The table description states that significant P values are printed in bold, but there are no P values in bold in the table.

Response:

We thank the reviewer for this comment and apologize for this oversight. Accordingly, we have revised Tables 3 and 4 (Pages 9-10).

5) Page 9 - row 274: Naive instead of naïve

Response:

We thank the reviewer for this comment. In accordance with the reviewer's advice, we have corrected the text (Page 9, line 282).

Reviewer 2 Report

Comments to the authors

In this study, Dr. Kuroda et al. compared the safety and efficacy of microwave ablation using ThermosphereTM technology versus radiofrequency ablation for hepatocellular carcinoma by A propensity score-matched analysis. The results of this study are clinically important. In addition, the manuscript is well written.

I have a comment as bellow.

  1. In the Methods section (2.1. Patients), authors described as follows:” The decision to perform MWA or RFA was made based on consensus between at least two hepatologists (HK and TO) who each had >15 years of experience. In addition, in the Table 1, the tumor size and the percentage of high-risk location met were significantly differences between MWA and RFA group in unmatched cohort. Please describe more detail regarding to the decision to perform MWA or RFA in the Methods section.

Author Response

RESPONSES TO REVIEWER 2

We thank the reviewer for the thoughtful and constructive assessment of our manuscript. We have addressed all concerns raised. Our responses to the reviewer’s comments are in detail as follows:

Please describe more detail regarding to the decision to perform MWA or RFA in the Methods section.

Response:

We thank the reviewer for this constructive comment. The decision to perform MWA or RFA was made based on consensus between at least two hepatologists who each had >15 years of experience. In most cases, there was no disagreement between the two hepatologists; in a few cases, their opinions differed, and a third hepatologist provided the decision. In the early stages of this study, we had more experience with RFA than with MWA. Therefore, MWA was chosen more carefully initially for HCC nodules at high-risk locations. In accordance with the reviewer's advice, we have revised the text (Page 3, lines 96-101).
